# Assessing plant phenological changes based on drivers of spring phenology

Yong Jiang[1], Stephen J Mayor[2], Xiuli Chu[3], Xiaoqi Ye[4], Rongzhou Man[2]*, Jing Tao[5]*, Qing-Lai Dang[6]

[1]Key Laboratory of Ecology of Rare and Endangered Species and Environmental Protection of Ministry of Education, Guangxi Normal University, Guilin, China; [2]Ontario Ministry of Natural Resources and Forestry, Ontario Forest Research Institute, Sault Ste. Marie, Canada; [3]Shanghai Botanical Garden, Shanghai Engineering Research Center of Sustainable Plant Innovation, Shanghai, China; [4]Research Institute of Subtropical Forestry, Chinese Academy of Forestry, Beijing, China; [5]Jilin Provincial Academy of Forestry Sciences, Changchun, China; [6]Faculty of Natural Resources Management, Lakehead University, Thunder Bay, Canada

*For correspondence:
rongzhou.man@ontario.ca (RM);
taojing8116@126.com (JT)

Competing interest: The authors declare that no competing interests exist.

## eLife Assessment

This study introduces a **valuable** new metric-phenological lag-to help partition the drivers of observed versus expected shifts in spring phenology under climate warming. The conceptual framework is clearly presented and supported by an extensive dataset, and the revisions have improved the manuscript, though some concerns—particularly regarding uncertainty quantification, spatial analysis, and modeling assumptions—remain only partially addressed. The strength of evidence is generally **solid**, but further analysis would help to validate the study's conclusions.

**Abstract** Understanding plant phenological responses to climate warming is essential for predicting shifts in plant communities and ecosystems. However, this remains challenging when sensitivity analyses overlook the underlying drivers of spring phenology. In this article, we present a new measure *phenological lag* to quantify the overall effect of phenological constraints, including insufficient winter chilling, photoperiod, and environmental stresses, based on observed response and that expected from species-specific changes in spring temperatures, that is, changes in spring forcing (degree days) from warming and average temperature at budburst with the warmer climate. We applied this new analytical framework to a global dataset with 980 species and 1527 responses to synthesize observed changes in spring budburst (leafing or flowering) and investigate the mechanisms of differential phenological responses reported previously. We found longer phenological lags with experimental studies and native plants in flowering, likely due to a more stressful environment associated with warmer and drier climate. Smaller forcing changes were mainly responsible for the smaller responses in leafing and flowering in the boreal region (compared to the temperate region) and in grass leafing (compared to trees and shrubs). Higher budburst temperatures also contributed to the smaller responses in flowering for experimental studies and with herbs and grasses. The effects of altitude, latitude, mean annual temperature (MAT), and average spring temperature change were minor (all combined <2.5% variations), while those of photoperiod and long-term precipitation were not significant in influencing spring phenology. Our method helps to determine mechanisms responsible for changes in spring phenology and differences in plant phenological responses.

## Introduction

Plant phenology, particularly in spring, is shifting with warming climate (*Fitter and Fitter, 2002*; *Post et al., 2018*; *Gallinat and Primack, 2016*). Numerous observational studies and controlled experiments, conducted over a range of climatic and phenological conditions (*Huang et al., 2020*; *Parmesan, 2007*; *Prevéy et al., 2019*; *Root et al., 2003*; *Wolkovich et al., 2012*), have produced varying results, with observed phenological changes differing by research approach (*Wolkovich et al., 2012*), climatic region (*Parmesan, 2007*; *Post et al., 2018*; *Prevéy et al., 2017*; *Zhang et al., 2015*), and functional group (*Parmesan, 2007*; *Root et al., 2003*; *Willis et al., 2010*; *Wolkovich et al., 2013*; *Zettlemoyer et al., 2019*). For example, observed changes are often smaller with experimental studies (*Wolkovich et al., 2012*), native species (*Wolkovich et al., 2013*), and warmer regions (*Prevéy et al., 2017*). Understanding these differences (also referred to as discrepancies, disparities, or mismatches) is crucial for predicting changes in plant communities and ecosystems (*Fitter and Fitter, 2002*; *Polgar et al., 2014*; *Gallinat and Primack, 2016*) but difficult using sensitivity analysis that is based on a relative rate of change, that is, change in days per degree Celsius or year/decade (*Ge et al., 2015*; *Parmesan, 2007*; *Wolkovich et al., 2012*).

Plants in the northern hemisphere require an accumulation of cool winter temperatures (winter chilling) to break dormancy and an accumulation of warm spring temperatures (spring forcing, degree days) to initiate budburst (e.g., leafing or flowering) (*Piao et al., 2019*; *Way and Montgomery, 2015*). If chilling requirements are fully met and other phenological constraints (e.g., photoperiod effect and environmental stresses) do not change with warming, the changes in spring phenology, in response to a warmer climate, result primarily from changes in spring temperatures, that is, changes in spring forcing and rate of forcing accumulation (i.e., temperature) at budburst with the warmer climate. That is, higher budburst temperatures are associated with smaller observed changes and sensitivity, opposite to the effects of forcing changes (*Chu et al., 2021*; *Chu et al., 2023*). As species often differ in timing of budburst, both forcing change and budburst temperature are species-specific, reflecting variations of individual species in phenological responses to climate warming (*Chu et al., 2021*; *Chu et al., 2023*). The effects of insufficient winter chilling, photoperiod (day length), and environmental stresses (e.g., drought and spring freezing) are similar in constraining the advance of spring phenology (*Chen et al., 2011*; *Huang et al., 2019*; *Körner and Basler, 2010*; *Ma et al., 2019*; *Way and Montgomery, 2015*). Therefore, the difference between observed phenological change and that expected from forcing change and budburst temperature represents phenological difference due to changes in phenological constraints and can be thereafter named phenological lag. Separating the effects of different constraints is possible if soil moisture or plant water status is monitored (*Huang et al., 2019*; *Post et al., 2022*), or if species-specific chilling and photoperiod effects (*Man et al., 2017a*; *Man et al., 2021a*; *Fu et al., 2019*) or cold hardiness (*Man et al., 2017b*; *Man et al., 2021b*) are known from previous research. For example, phenological lag can represent chilling effects when photoperiod effect and environmental stresses remain unchanged with warming (*Chu et al., 2021*).

The changes in spring phenology are often assessed by sensitivity, a parameter that does not account for uneven warming patterns and responses (*Beaubien and Hamann, 2011*; *Keenan et al., 2020*; *Post et al., 2018*; *Rafferty et al., 2020*) and therefore does not provide meaningful insights into the magnitude and underlying mechanisms of changes in spring phenology (*Chu et al., 2021*; *Chu et al., 2023*). For example, the same average temperature change (spring or annual temperature) can lead to different changes in spring phenology if warming occurs unevenly before and after budburst. Under this circumstance, interpretations based on sensitivity analysis can be misleading. Here, we present a new method to partition observed phenological changes based on changes in spring phenology and temperatures with control (baseline) and warmer climates. We applied this analytical approach to meta-analyze phenological changes reported in the literature and investigated the mechanisms of the differential responses reported previously on research approach (observational or experimental), species origin (native or exotic), climatic region (boreal or temperate), and growth form (tree, shrub, herb, or grass). Our objective was to determine how phenological responses differ among different groups and how differential responses are related to the drivers of spring phenology, that is, forcing change, budburst temperature, and phenological lag.

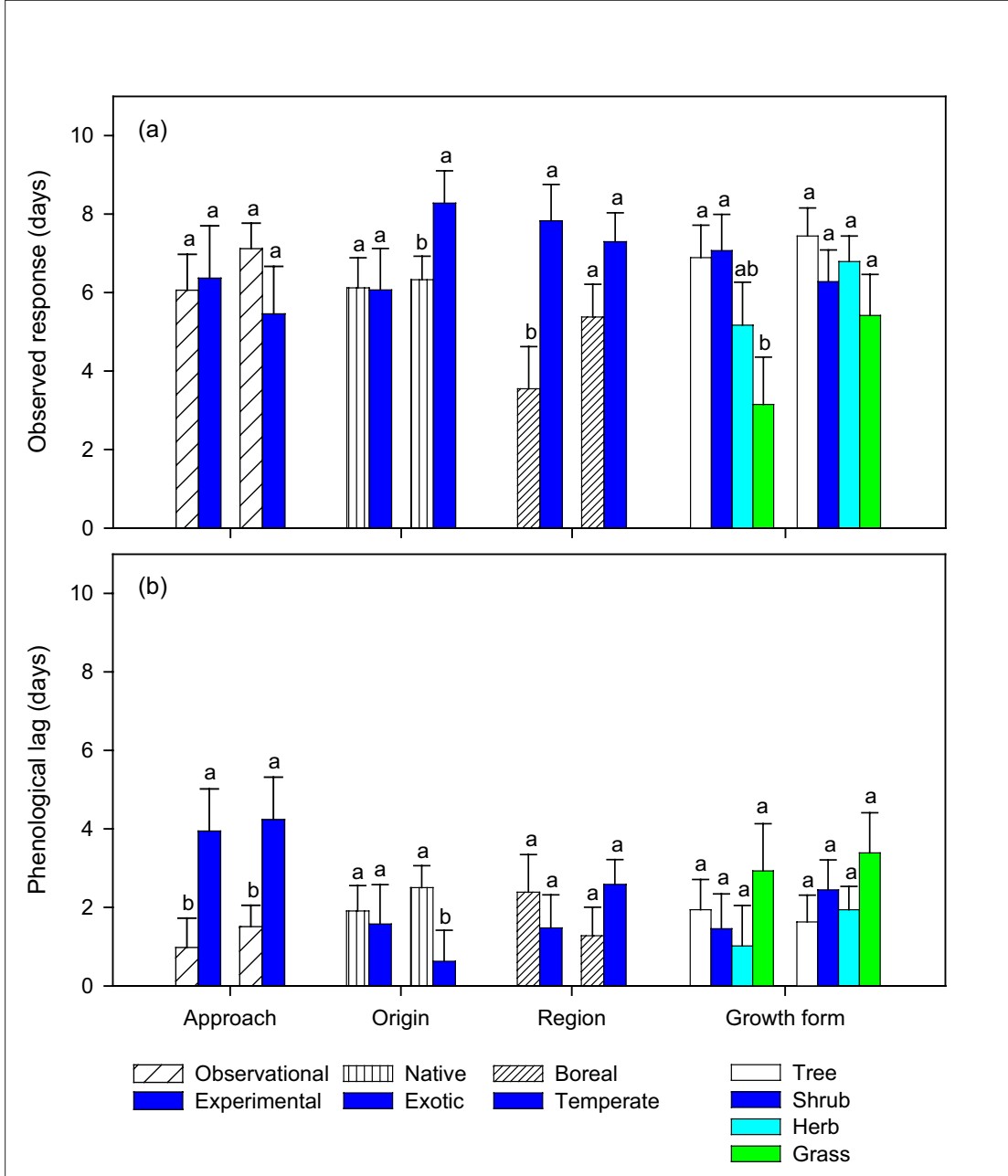

**Figure 1.** Observed responses and phenological lags (least square means ± standard errors) in leafing and flowering (left and right set of bars in each pair, respectively) by research approach (observational and experimental), species origin (native and exotic), climatic region (boreal and temperate), and growth form (tree, shrub, herb, and grass) extracted from reported plant phenological changes in spring. Phenological lags are calculated from the differences between observed responses and those expected from species-specific changes in spring temperatures. Different letters indicate means that differ significantly ($P<0.05$).

## Results

Across 66 studies with 980 species and 1527 responses, observed responses ($N_O$) ranged from −23.8 to +26.8 days for leafing and −36.3 to +49.9 days for flowering, with an average of +6.0 days for both events (*Supplementary file 1C*). These changes differed from expected responses ($N_E$) which ranged from −3 to +27 days for leafing and −3 to +26 days for flowering and averaged +7.8 and +9.5 days, respectively. Observed phenological responses were thus smaller than expected by 1.8 and 3.5 days, on average, for leafing and flowering, respectively.

**Table 1.** Mean values of climatic, phenological, and biological variables by budburst (leafing or flowering), research approach (observational or experimental), species origin (native or exotic), climatic region (boreal or temperate), and growth form (tree, shrub, herb, or grass).

Budburst temperature = average temperature at budburst under the warmer climate, forcing change = warming-induced changes in spring forcing prior to budburst, spring warming = average spring temperature change, and phenological lag = difference between observed response and that expected from forcing change and budburst temperature.

| Budburst | Spring phenology (Julian) | Budburst temperature (°C) | Altitude (m) | Latitude (°N) | MAT (°C) | Forcing change (degree days) | Spring warming (°C) | Phenological lag (day) | Sample size (N) |
|---|---|---|---|---|---|---|---|---|---|
| Leafing | 111 | 11.1 | 687 | 42.8 | 7.8 | 85 | 1.3 | 1.8 | 447 |
| Observational | 106 | 11.0 | 487 | 42.6 | 8.4 | 79 | 1.1 | 1.2 | 359 |
| Experimental | 130 | 11.2 | 1500 | 43.8 | 5.6 | 106 | 1.8 | 4.1 | 88 |
| Native | 113 | 11.0 | 744 | 43.3 | 7.4 | 85 | 1.3 | 1.9 | 379 |
| Exotic | 100 | 11.7 | 366 | 40.0 | 10.2 | 78 | 1.2 | 1.1 | 68 |
| Boreal | 128 | 10.2 | 1147 | 45.7 | 2.0 | 65 | 1.3 | 2.4 | 154 |
| Temperate | 102 | 11.5 | 444 | 41.3 | 10.9 | 94 | 1.3 | 1.5 | 293 |
| Tree | 105 | 11.7 | 364 | 41.9 | 10.1 | 101 | 1.4 | 1.9 | 246 |
| Shrub | 109 | 10.9 | 433 | 43.5 | 7.8 | 80 | 1.4 | 1.1 | 107 |
| Herb | 131 | 10.0 | 1955 | 43.3 | 1.8 | 51 | 0.8 | 1.2 | 54 |
| Grass | 123 | 8.9 | 1637 | 45.9 | 2.1 | 37 | 0.7 | 3.6 | 40 |
| Flowering | 138 | 13.7 | 534 | 45.5 | 9.0 | 126 | 1.3 | 3.5 | 1080 |
| Observational | 136 | 13.4 | 454 | 46.0 | 9.2 | 120 | 1.2 | 3.3 | 956 |
| Experimental | 154 | 16.7 | 1153 | 41.8 | 7.3 | 174 | 1.8 | 4.7 | 124 |
| Native | 142 | 13.8 | 617 | 45.9 | 8.5 | 124 | 1.3 | 3.7 | 850 |
| Exotic | 125 | 13.5 | 229 | 44.3 | 10.9 | 135 | 1.4 | 2.4 | 230 |
| Boreal | 167 | 14.6 | 1998 | 42.9 | 3.3 | 109 | 1.4 | 1.3 | 175 |
| Temperate | 133 | 13.6 | 251 | 46.0 | 10.1 | 130 | 1.3 | 3.9 | 905 |
| Tree | 114 | 12.7 | 289 | 43.5 | 10.5 | 119 | 1.5 | 1.4 | 183 |
| Shrub | 125 | 12.3 | 446 | 44.8 | 8.5 | 103 | 1.3 | 2.9 | 148 |
| Herb | 146 | 14.2 | 608 | 46.1 | 8.7 | 129 | 1.4 | 3.8 | 659 |
| Grass | 150 | 15.3 | 636 | 46.5 | 8.9 | 160 | 1.4 | 5.8 | 90 |

Note: effects of forcing change and budburst temperature illustrated in Partitioning observed changes.

a) the difference in forcing change between boreal and temperate regions for leafing is 29 (94 – 65), equivalent to 2.8 day expected responses under the average budburst temperature of boreal region (10.2). Similarly, the difference in flowering is equivalent to 1.4 day expected responses, i.e., 21/14.6.

b) the difference in forcing change between grasses and trees + shrubs for leafing is 53 ((101+80)/2 – 37), equivalent to 6.0 day expected responses under the average budburst temperature of grasses (8.9).

c) the difference in forcing change between experimental and observational studies for leafing is 27 (106 –79), equivalent to 2.5 day expected responses under the average budburst temperature of observational studies (11.0).

d) different budburst temperatures explain 2.6 day difference in flowering responses between experimental and observational studies, i.e., 174/13.4 – 174/16.7, and 1.8 day difference in flowering responses between woody (trees and shrubs) and non-woody (herbs and grasses) plants, i.e., (129+160)/(12.7+12.3) – (129+160)/(14.2+15.3).

In experimental studies, observed flowering response was non-significantly smaller than those from observational studies (1.7 days, $P=0.232$, *Figure 1a*), while phenological lag was significantly longer for both leafing (3.0 days, $P=0.028$) and flowering (2.7 days, $P=0.027$) (*Figure 1b*). Observed flowering response in exotic species was 2 days greater than in native species ($P=0.016$), likely due to differences in phenological lag (1.9 days, $P=0.019$). Observed leafing and flowering responses in the boreal region were 4.3 and 1.9 days smaller ($P=0.004$, $P=0.088$), respectively, than in the temperate region,

**Table 2.** Final stepwise regression coefficients and variable influence on observed phenological responses using Akaike information criterion (AIC).

| Budburst | Intercept | Budburst temperature | Altitude | Latitude | MAT | Forcing change | Spring warming | Phenological lag | R² sample (N) |
|---|---|---|---|---|---|---|---|---|---|
| Leafing | 2.6439 | –0.6075 | 0.0003 | 0.0732 | 0.1480 | 0.0736 | 0.9463 | –0.9738 | 0.9453 |
| AIC (%) | 449 (27.7) | 340 (20.9) | 2 (0.1) | 15 (0.9) | 11 (0.7) | 68 (4.2) | 7 (0.4) | 732 (45.1) | (447) |
| Flowering | 4.5210 | –0.5917 | 0.0003 | 0.0869 | 0.1466 | 0.0521 | 0.7620 | –0.9893 | 0.9653 |
| AIC (%) | 785 (18.1) | 247 (5.7) | 7 (0.2) | 20 (0.5) | 32 (0.7) | 826 (19.1) | 5 (0.1) | 2409 (55.6) | (1080) |

Note: Budburst temperature (°C) = average temperature at budburst (leafing or flowering) with the warmer climate.

MAT (°C) = long-term mean annual temperature.

Forcing change = warming-induced changes in spring forcing prior to budburst.

Spring warming (°C) = average spring temperature change (Julian day 1 to 182).

Phenological lag (days) = difference between observed response and that expected from forcing change and budburst temperature.

Spring phenology (date of budburst with the baseline climate; Julian days) and long-term mean precipitation (MAP, mm) were excluded from final models due to lack of significance.

Forcing change, budburst temperature, and phenological lag are calculated from phenological and temperature data for individual species, spring phenology and observed responses are provided in the original studies, and spring warming and long-term MAT and MAP are provided in the original studies, calculated from baseline temperatures, or extracted from Google Maps, FreeMapTools, or WorldClim Database (**Fick and Hijmans, 2017**).

partially due to smaller forcing changes (equivalent to 2.8 days in leafing and 1.4 days in flowering based on forcing changes and budburst temperatures, *Table 1* note a). Observed leafing response in grasses was 3.8 days smaller than in trees and shrubs (*P*=0.033), likely due to differences in forcing changes (6.0 days, *Table 1* note b).

Other than phenological lag, forcing change and budburst temperature were the most influential variables affecting observed responses (*Table 2*). The remaining variables combined (altitude, latitude, MAT, and spring warming) explained <2.5% variations in observed leafing and flowering responses.

## Discussion
### Observational vs. experimental

By partitioning observed changes based on drivers of spring phenology, we clarified some of the uncertainty regarding the underlying mechanisms of differences in plant phenological response to climate warming. By synthesizing global data, we found that observed flowering responses in experimental studies were non-significantly smaller than that in observational studies, but this was not the case for leafing responses; these findings differ from those based on temperature sensitivity (*Wolkovich et al., 2012*). The greater warming in experimental studies did not produce greater observed responses, due to longer phenological lags, which may have resulted in the reduced sensitivity (*Wolkovich et al., 2012*). First, experimental studies are often conducted at higher altitude sites with lower MAT, later spring phenology, and therefore longer accumulation of winter chilling. The longer phenological lags are therefore unlikely the result of insufficient winter chilling. Second, artificial warming has typically provided higher forcing changes and budburst temperatures than natural climate warming (see *Table 1*). Greater forcing changes produce larger expected responses (observed response + phenological lag), whereas higher budburst temperatures reduce expected responses (*Chu et al., 2021*; *Prevéy et al., 2017*; *Wolkovich et al., 2021*). The high temperatures in the artificial warming structures (*Marion et al., 1997*) can also reduce humidity and soil moisture (*Ettinger et al., 2019*; *Huang et al., 2019*), slowing spring development (*Ganjurjav et al., 2020*; *Huang et al., 2019*; *Moore et al., 2015*), particularly in early spring when temperatures in the surrounding environment are low, thus restricting soil water movement. While the different forcing changes adequately account for the differences in expected leafing responses (2.5 out of 3.3 days, *Table 1* note c), the large differences in forcing changes but similar expected responses in flowering (8.63 and 9.07 days, *Figure 1*) suggests a strong effect of budburst temperature (*Table 1* note d) (*Chu et al., 2023*). Thus, warming-induced moisture stress and high budburst temperatures may interact to alter observed phenological responses and temperature sensitivity in experimental studies (*Wolkovich et al., 2012*). Future

experiments should consider humidity, soil moisture, and plant water status to better understand disparities between observational and experimental studies and relate plant phenological changes from experimental warming to those from natural climate change (*Wolkovich et al., 2012*).

## Native vs. exotic species

Exotic species are often reported with more noticeable observed responses to warming, a trend that is generally based on flowering (*Calinger et al., 2013*; *Willis et al., 2010*; *Wolkovich et al., 2013*). The ecological implications of this response are well described (*Polgar et al., 2014*; *Zettlemoyer et al., 2019*), but the underlying mechanisms are not well understood. Lower budburst temperatures faced by early-start exotics are likely partially the cause (*Chu et al., 2021*; *Chu et al., 2023*), as shown by a reverse trend for late-start exotics (*Zohner and Renner, 2014*). In this synthesis, phenological lag did not increase from early leafing to late flowering for exotic species, a trend that is consistent with dormancy release commonly reported on leaf and flower buds (*Campoy et al., 2013*; *Gariglio et al., 2006*; *Hussain et al., 2015*; *Wall et al., 2008*), contrary to that for native species (*Figure 1b*). The smaller observed response in flowering for native species is associated with longer phenological lags and accumulations of winter chilling (*Table 1*), suggesting more stressful environments later in the season or higher stress sensitivity with reproductive events. Comparatively, plants starting growth early in the growing season may be less restricted by soil moisture that often recharges from winter precipitation and is depleted with increased moisture consumption over time (*Sherry et al., 2007*; *Wolkovich et al., 2013*; *Zettlemoyer et al., 2019*), particularly in dry climates (*Man and Greenway, 2013*). Moisture stress can progressively develop over the season (*Piao et al., 2019*; *Wolkovich et al., 2013*; *Zettlemoyer et al., 2019*) and differentially affect exotic and native species with differing spring phenology (*Stuble et al., 2021*; *Willis et al., 2010*; *Wolkovich et al., 2013*). The smaller observed responses in flowering of native species suggest reproductive disadvantage (*IPCC, 2014*; *Zettlemoyer et al., 2019*). The consistent leafing response, however, does not support suggestions that late-start native species would have relatively shorter active growing seasons and, therefore, a competitive growth disadvantage relative to early-start exotic species with climate warming (*Fitter and Fitter, 2002*; *Morin et al., 2009*; *Gallinat and Primack, 2016*). Total thermal benefits from climate warming do not differ among species with differing spring phenology (*Chu et al., 2021*). Given the projected increases in temperature and decreases in precipitation for many parts of the world (*IPCC, 2014*), future studies should assess the differences in drought sensitivity between vegetative and reproductive events and among functional groups and climatic regions.

## Boreal vs. temperate region

Spatial variations along altitudinal and latitudinal gradients are often reported but not well understood (*Ge et al., 2015*; *Morin et al., 2009*; *Parmesan, 2007*; *Zhang et al., 2015*). Our analysis indicates smaller observed responses in the boreal region (MAT<6°C) due to less forcing changes (*Table 1*), but not to longer phenological lag (*Figure 1*). The similar spring warming but less forcing change suggests that the temperature increases in the boreal region occur more frequently in the winter (*Beaubien and Hamann, 2011*; *Shen et al., 2015*) when temperatures are below freezing and don't contribute to spring forcing (*Man and Lu, 2010*). The uneven warming has been reported in both natural and experimental settings (*Beaubien and Hamann, 2011*; *Prevéy et al., 2017*; *Shen et al., 2015*; *Slaney et al., 2007*; *Yang et al., 2020*) and supports the claim that average temperature changes do not represent species-specific forcing changes (*Chu et al., 2021*; *Keenan et al., 2020*; *Wolkovich et al., 2021*). When forcing changes are held constant, however, observed responses increase with altitude and latitude, as shown by the positive regression coefficients in both leafing and flowering models (*Table 2*). The possible mechanisms behind this may be lower budburst temperature and less moisture stress at higher altitude and latitude (*Rafferty et al., 2020*), as shown by the negative collinearity of altitude and latitude with budburst temperature and phenological lag in both leafing and flowering (data not shown). Therefore, the boreal region, depending on forcing changes, can have smaller (*Ge et al., 2015*; *Menzel et al., 2006*; *Rafferty et al., 2020*; *Shen et al., 2015*) or greater (*Ge et al., 2015*; *Parmesan, 2007*; *Post et al., 2018*; *Prevéy et al., 2017*) phenological responses, spatial trends that may be different from those suggested by phenological sensitivity (*Liu et al., 2019*; *Post et al., 2018*). Consequently, climate change may not result in general phenological convergence across altitudes

and latitudes (*Rafferty et al., 2020*; *Shen et al., 2015*; *Tao et al., 2021*), contrary to suggestions by others based on small scale studies (*Prevéy et al., 2017*; *Vitasse et al., 2018*; *Ziello et al., 2009*).

## Trees, shrubs, herbs, and grasses

Smaller observed responses in herb and grass leafing support findings from some early syntheses (*König et al., 2018*; *Parmesan, 2007*), but not those that show a greater response from non-woody plants (*Ge et al., 2015*; *Post and Stenseth, 1999*; *Root et al., 2003*) or no differences (*Stuble et al., 2021*). Herbs and grasses often resume growth earlier than woody shrubs and trees (*Badeck et al., 2004*; *Heberling et al., 2019*) and should have lower budburst temperatures and therefore greater phenological responses to climate warming (*Chu et al., 2021*; *Prevéy et al., 2017*; *Wolkovich et al., 2021*). However, different growth forms are often studied separately on sites in different climatic regions, making comparisons challenging. Furthermore, most studies on herb and grass leafing are conducted in boreal regions at high altitudes, low MAT, late spring phenology, low forcing changes, and low spring warming (*Table 1*); the extremely small forcing changes resulted in small observed responses despite lower budburst temperatures (*Figure 1a*). The observed responses with herbs and grasses would not be smaller if converted to temperature sensitivity (see *Table 1* for large differences in spring warming for leafing). In contrast to leafing, herb and grass flowering studies have greater forcing changes and higher budburst temperatures (*Table 1*), the latter would lower both expected and observed responses (*Table 1* note d) (*Chu et al., 2021*; *Prevéy et al., 2017*; *Wolkovich et al., 2021*) and potentially induce moisture stress and increase phenological lags (*Ganjurjav et al., 2020*; *Huang et al., 2019*; *Moore et al., 2015*; *Post et al., 2022*), particularly for grasses (*Sherry et al., 2007*; *Zettlemoyer et al., 2019*) that have shallower root systems (*Kulmatiski and Beard, 2013*; *Schenk and Jackson, 2002*).

## Mechanistic assessment of changes in spring phenology

Averaged across all observational and experimental studies, observed responses and phenological lags are both positive, suggesting that climate warming advances spring phenology but not at rates expected from changes in spring temperatures. The positive lags are likely due to more stressful environments with warmer and drier climate (*Huang et al., 2019*; *Sherry et al., 2007*; *Zettlemoyer et al., 2019*). We quantified phenological lags from changes in spring phenology and temperatures, an approach that does not require species-specific chilling or forcing needs that are often unavailable (*Fitter and Fitter, 2002*; *Morin et al., 2009*) or variable in methods of estimation across species and studies (*Man and Lu, 2010*; *Ettinger et al., 2020*; *Zhang et al., 2018*). The relatively early stage of climate warming and association of longer phenological lags with longer accumulation of winter chilling (*Table 1*) suggests that current climate warming is not likely to induce a general chilling shortage (*Chu et al., 2023*; *Ettinger et al., 2020*; *Tao et al., 2021*; *Yang et al., 2020*), although incidental effects on individual species or in particular conditions can occur (*Fu et al., 2015*). Similarly, the variable and minor effects of photoperiod (*Basler and Körner, 2012*; *Chuine et al., 2010*; *Ettinger et al., 2020*; *Zohner et al., 2016*) (insignificant spring phenology; see *Table 2*) often reported in studies with extreme warming scenarios (*Caffarra and Donnelly, 2011*; *Fu et al., 2019*; *Laube et al., 2014*) or compounded with that of budburst temperatures (*Chu et al., 2021*), minor effects by nutrient availability (*Piao et al., 2019*), or incidental spring freezing (*Man et al., 2009*; *Man et al., 2021b*) are not likely to cause the systemic differences in phenological lag between observational and experimental studies or between native and exotic species.

Our analysis indicates that both forcing change (quantity) and budburst temperature (rate of forcing accumulation) strongly influence the magnitude of change in spring phenology, consistent with the findings by *Chu et al., 2021* that smaller phenological response with late season species is largely due to their higher budburst temperatures. In phenological research, the influence of budburst temperature is not adequately recognized (*Chu et al., 2021*; *Chu et al., 2023*) and can be confounded with progressive increases of phenological lag with greater chilling requirements or insufficient winter chilling (*Asse et al., 2018*; *Morin et al., 2009*; *Gallinat and Primack, 2016*), photoperiod constraints (*Körner and Basler, 2010*; *Shen et al., 2014*; *Way and Montgomery, 2015*), or moisture stress (*Piao et al., 2019*; *Wolkovich et al., 2013*; *Zettlemoyer et al., 2019*) suggested for late season species. The difference in the significance of budburst temperatures between leafing and flowering models (*Table 2*) suggests that the influence of budburst temperatures can be greater if budburst

temperatures decrease with the advance of spring phenology and progress of climate warming (*Tao et al., 2021*). Compared to spring average temperature change that explained <0.5% of the variation in observed responses (*Table 2*), forcing change quantifies climate warming effect relevant to phenological change of individual species and reduces uneven warming effects among species with different spring phenology (*Chu et al., 2021*) and areas in different climatic regions (*Post et al., 2018*; *Yang et al., 2020*). Both forcing change and budburst temperature can be readily extracted from phenological and temperature data, as demonstrated in this synthesis, but have not been used in assessment of plant phenological changes in spring.

## Conclusions and caveats

In this article, we outline an analytical framework to partition phenological changes based on drivers of spring phenology and report differential phenological responses identified through the meta-analysis of observed changes and phenological lag. Longer phenological lag, likely resulting from more stressful environments with warmer and drier climate, helped to explain smaller responses with experimental studies and native plants in flowering, while less forcing changes were mainly responsible for the smaller responses in leafing and flowering in the boreal region and in grass leafing. Some of these differential responses are different from those reported previously based on sensitivity analysis.

Our approach does not require species-specific chilling and forcing needs, chilling–forcing relationships, or temperature response models that are often unavailable or variable in methods of estimation across species and studies. As chilling and forcing models reflect physiological aspects of chilling and forcing processes, the use of alternative base temperatures or forcing models would not change the partitioning of phenological changes for individual species, but would affect comparisons among species under different temperature regimes, i.e., different climatic regions or timing of spring phenology.

While our method helps to understand changes in spring phenology and differences in plant phenological responses across different functional groups or climatic regions, the ecological implications of phenological lag can be uncertain without investigation of individual constraints. In this synthesis, the effects of photoperiod and insufficient winter chilling are likely limited across all studies with the current level of climate change. In the boreal region with a long winter, insufficient winter chilling is unlikely to occur with the levels of climate warming projected (*Chu et al., 2021*; *Tao et al., 2021*). Phenological lag reflects the overall lag effect and highlights the need to investigate individual constraints that can be specifically determined at individual study level if biological and environmental constraints are known from on-site monitoring or previous research.

## Materials and methods

### Partitioning observed changes

Species-specific forcing change ($F_C$) can be calculated from the difference in degree days (>0°C, see *Man and Lu, 2010*) between baseline (or control) ($\sum_1^i T_{Ci}$) and warmer ($\sum_1^i T_{Wi}$) climates at budburst (leafing or flowering) with the baseline climate ($O_{Ci}$) (see *Figure 2*).

$$F_C = \sum_1^i T_{Wi} - \sum_1^i T_{Ci}$$

The phenological response expected under the null hypothesis that climate warming does not alter phenological constraints can be estimated from forcing change and species phenology with the baseline climate (*Figure 2*):

$$N_{Ei} = \min \left| \sum_1^{i-n} T_{Wi} - \sum_1^i T_{Ci} \right|$$

where $N_E$ is the difference in the number of days (n) between control and warmer climates in reaching species forcing threshold ($\sum_1^i T_{Ci}$ at $O_{Ci}$). Under this hypothesis, spring phenology is expected to shift from $O_{Ci}$ with the baseline climate to $E_{Wi}$ with the warmer climate. $N_E$ is typically positive, meaning

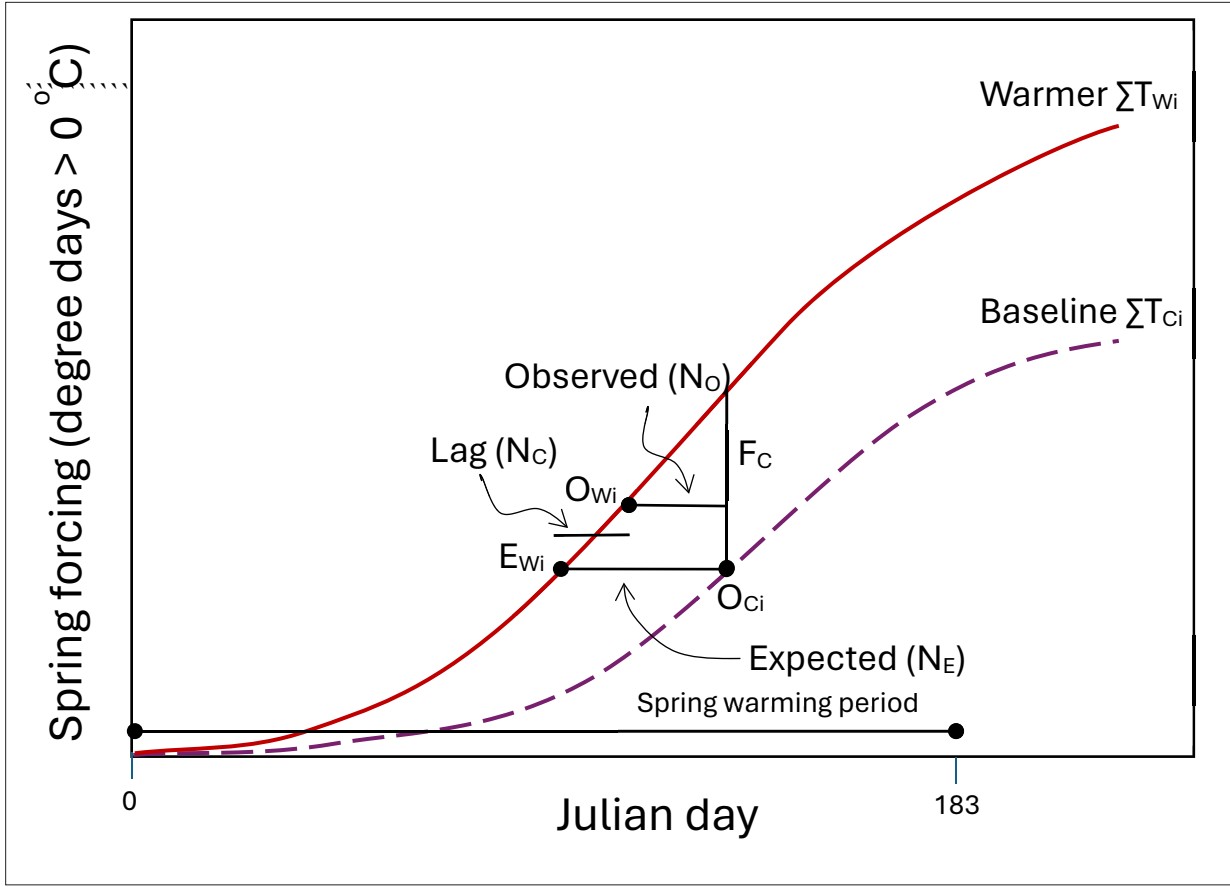

**Figure 2.** A diagram showing the relationships among forcing change ($F_C$, difference in spring forcing (degree days) between baseline $\sum_1^i T_{Ci}$ and warmer $\sum_1^i T_{Wi}$ climates at budburst with the baseline climate $O_{Ci}$), expected response ($N_E$, difference between baseline and warmer climates in reaching species forcing threshold, i.e., $\sum_1^i T_{Ci}$ at $O_{Ci}$), budburst temperature ($F_C/N_E$, average temperature or rate of forcing accumulation at budburst with the warmer climate), and phenological lag ($N_C$, difference between expected $N_E$ and observed $N_O$ responses or between expected $E_{Wi}$ and observed $O_{Wi}$ phenology with the warmer climate).

earlier leafing and flowering, but can be negative if temperatures decrease over time in observational studies.

Budburst temperature ($T_B$) is the average temperature or rate of forcing accumulation within the window of expected phenological response (between $O_{Ci}$ and $E_{Wi}$) and can be determined from forcing change and expected response.

$$T_B = F_C/N_E$$

The difference between expected ($N_E$) and observed ($N_O$) responses is phenological lag ($N_C$) reflecting phenological difference due to change in phenological constraints induced by warming.

$$N_C = N_E - N_O$$

$N_C > 0$ indicates increasing phenological constraints that may result from increasing chilling deficiency, photoperiod restriction, moisture stress, or risks of spring frosts; $N_C = 0$ indicates no changes; and $N_C < 0$ indicates decreasing constraints with warmer climate.

## Phenological data

We followed the guidelines of PRISMA (Preferred Reporting Items for Systematic Reviews and Meta-Analyses) (*O'Dea et al., 2021*; *Page et al., 2021*) to build up the dataset on plant spring phenological response to both experimental and climatic warming. We used Web of Science and Google Scholar

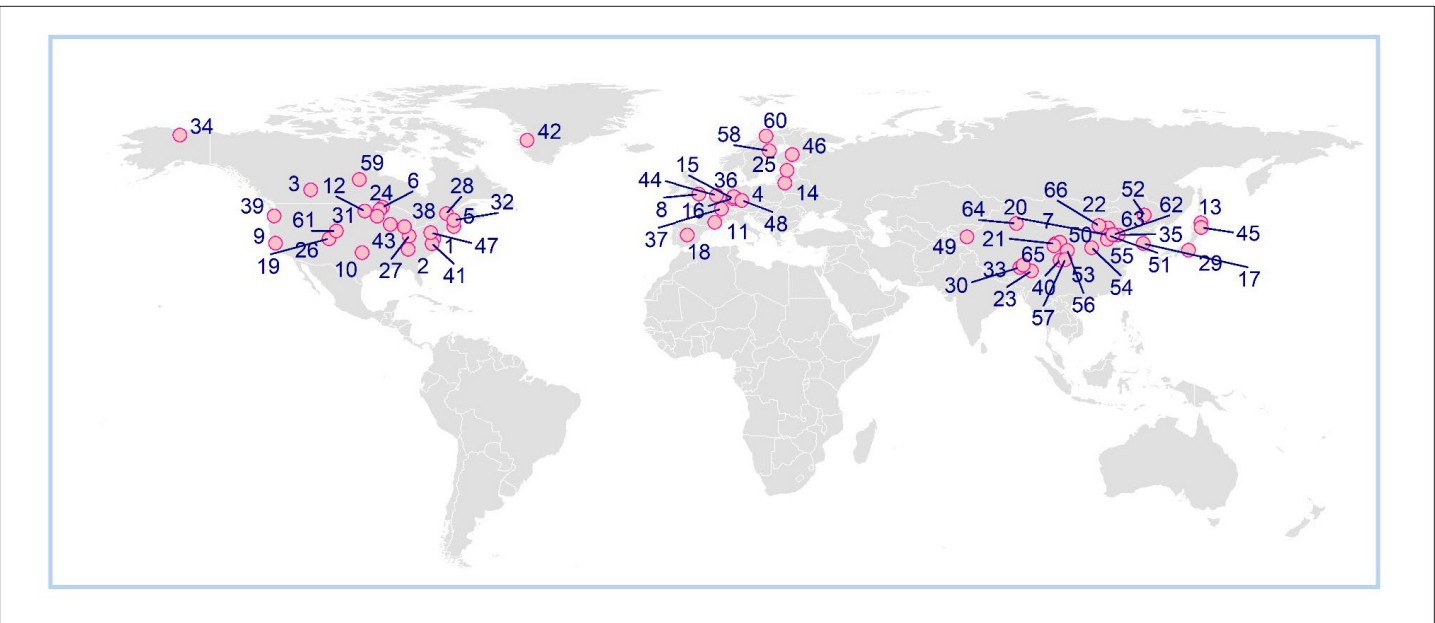

**Figure 3.** Distribution of phenological studies used in this synthesis. Numbers correspond to study information in *Supplementary file 1A*. For studies reporting average phenological responses for multiple locations, geographic centers are shown.

to search experimental and observational studies on warming-induced changes in spring phenology. The following criteria were used to select papers: (a) accessible peer-reviewed articles published in scientific journals for boreal (MAT<6°C) and temperate (MAT ≥6°C) regions in the northern hemisphere, (b) studies reporting spring budburst (leafing or flowering) before Julian day 213 (July 31), (c) studies containing different climatic conditions of baseline (lower temperatures in early period of observations or control treatment) and warmer (higher temperatures in later periods of observations or warmer treatment) climates, (d) access to local temperature data (only one study was excluded due to lack of nearby temperature data within 50 km), and (e) reported phenological changes in unique locations, species, and periods of observations/warmer treatments. When publications did not contain sufficient phenological information, we accessed data directly from phenological databases (e.g., China's National Earth System Science Data Center). When results were reported in graphical format, we extracted the required data using the distance measuring tool in Adobe Acrobat Reader DC. In experimental studies with multiple warming treatments, we selected the treatment with the smallest temperature increases to represent a likely scenario of climate change and to maintain data independence. We also excluded studies where the timing of spring budburst, total temperature change, or observational time periods were not available.

In total, 66 studies were identified from 87 locations, containing 1527 phenological responses for 980 species (*Figure 3*; *Supplementary file 1A–C*). For studies reporting multiple phenological stages, we only used data for the first occurrence of the events. To account for variation in phenological responses, each study location was characterized by research approach (observational or experimental), climatic region (boreal or temperate), biogeographic origin of species (exotic or native to study area), and growth form (tree, shrub, herb, or grass).

## Climate data

Baseline and warmer temperature data for calculating forcing change, expected response, budburst temperature, phenological lag, spring warming (average temperature change from Julian day 1–182), and long-term MAT and MAP were obtained from 9 different sources (*Supplementary file 1D*). When temperature data were available from multiple sources, we selected the data set with the fewest missing values. For each phenological response, the number of weather stations within 50 km of data collection ranged from 1 for studies at a single location to 14 for observational studies across large geographic areas (*Supplementary file 1B*).

## Data analysis

We calculated forcing change, expected response, budburst temperature, and phenological lag for each of the 1527 responses compiled. We used linear mixed-effects models to explore the differences in observed responses and phenological lags between research approaches (observational vs. experimental), species origins (native vs. exotic), and climatic regions (boreal vs. temperate), and among growth forms (trees, shrubs, herbs, and grasses). Observed response is the total phenological change observed (commonly called phenological change or response), while phenological lag represents the lag effect of all constraints and therefore modification of phenological change from the expectation based on changes in spring temperatures. A separate analysis was conducted for each of the four fixed effects and two budburst events (leafing and flowering). Location and species were treated as random effects.

To assess the influences of climatic, phenological, and biological variables, we used a stepwise regression with automated combined forward and backward selection by Akaike information criterion (AIC) (*Burnham and Anderson, 2002*) to select the best combination of variables for predicting observed responses ($N_O$).

$$N_O \sim A_L + L_A + MAT + MAP + D_T + T_B + F_C + S_W + N_C$$

The variables included were altitude ($A_L$), latitude ($L_A$), MAT, MAP, spring phenology ($D_T$), budburst temperature ($T_B$), forcing change ($F_C$), spring warming ($S_W$), and phenological lag ($N_C$), and not standardized prior to regression analysis.

## Supporting information

Temperature data and R codes for calculating forcing changes, expected responses, budburst temperature, spring warming, and phenological lags of individual studies are deposited at Dryad (https://doi.org/10.5061/dryad.dncjsxm9x).

# Acknowledgements

This work was supported by the Ontario Ministry of Natural Resources and Forestry (Canada), Guangxi Normal University (China), Jilin Provincial Academy of Forestry Sciences (China), Shanghai Botanical Garden (China), Research Institute of Subtropical Forestry (China), and Lakehead University (Canada). Lisa Buse, Gillian Muir, and Hasanki Gamhewa of Ontario Ministry of Natural Resources and Forestry and three anonymous reviewers provided constructive suggestions for improving earlier drafts of the manuscript. Phenological and temperature data provided by China's National Earth System Science Data Center are greatly appreciated.

# Additional information

### Funding

No external funding was received for this work.

### Author contributions

Yong Jiang, Xiaoqi Ye, Qing-Lai Dang, Data curation, Formal analysis, Writing – original draft, Writing – review and editing; Stephen J Mayor, Data curation, Formal analysis, Visualization, Writing – original draft, Writing – review and editing; Xiuli Chu, Data curation, Writing – original draft, Writing – review and editing; Rongzhou Man, Conceptualization, Resources, Data curation, Formal analysis, Supervision, Visualization, Methodology, Writing – original draft, Project administration, Writing – review and editing; Jing Tao, Conceptualization, Data curation, Methodology, Writing – original draft, Writing – review and editing

### Author ORCIDs

Rongzhou Man https://orcid.org/0000-0003-4560-5620

Reviewer #3 (Public review): https://doi.org/10.7554/eLife.106655.4.sa1

Author response https://doi.org/10.7554/eLife.106655.4.sa2

## Additional files

### Supplementary files
Supplementary file 1. A global dataset of observational and experimental studies. (**A**) A list of references for all studies included. (**B**) A summary of study information. (**C**) MetaData containing all data used in the analysis. (**D**) Sources of weather data.

MDAR checklist

### Data availability
Temperature data and R codes for calculating forcing changes, expected responses, budburst temperature, spring warming, and phenological lags of individual studies are deposited at Dryad (https://doi.org/10.5061/dryad.dncjsxm9x).

The following dataset was generated:

| Author(s) | Year | Dataset title | Dataset URL | Database and Identifier |
|---|---|---|---|---|
| Man R, Jiang Y, Mayor S, Chu X, Ye X, Tao J, Dang Q | 2025 | Assessing plant phenological changes based on drivers of spring phenology | https://doi.org/10.5061/dryad.dncjsxm9x | Dryad Digital Repository, 10.5061/dryad.dncjsxm9x |

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
