## [Editor Report · eLife Assessment]

This study introduces a **valuable** new metric-phenological lag-to help partition the drivers of observed versus expected shifts in spring phenology under climate warming. The conceptual framework is clearly presented and supported by an extensive dataset, and the revisions have improved the manuscript, though some concerns—particularly regarding uncertainty quantification, spatial analysis, and modeling assumptions—remain only partially addressed. The strength of evidence is generally **solid**, but further analysis would help to validate the study's conclusions.

---

## [Referee Report · Reviewer #3 (Public review)]

Summary:

The authors developed a new phenological lag metric and applied this analytical framework to a global dataset to synthesize shifts in spring phenology and assess how abiotic constraints influence spring phenology.

Strengths:

The dataset developed in this study is extensive, and the phenological lag metric is valuable.

Weaknesses:

The stability of the method used to calculate forcing requirements needs improvement, for example, by including different base temperature thresholds. In addition, the visualization of the results should be improved.

---

## [Author Response]

The following is the authors’ response to the previous reviews

**Reviewer #1 (Public review):**
Jiang et al. present a measure of phenological lag by quantifying the effects of abiotic constraints on the differences between observed and expected phenological changes, using a combination of previously published phenology change data for 980 species, and associated climate data for study sites. They found that, across all samples, observed phenological responses to climate warming were smaller than expected responses for both leafing and flowering spring events. They also show that data from experimental studies included in their analysis exhibited increased phenological lag compared to observational studies, possibly as a result of reduced sensitivity to climatic changes. Furthermore, the authors present evidence that spatial trends in phenological responses to warming may differ than what would be expected from phenological sensitivity, due to the seasonal timing of when warming occurs. Thus, climate change may not result in geographic convergences of phenological responses. This study presents an interesting way to separate the individual effects of climate change and other abiotic changes on the phenological responses across sites and species.Strengths:A straightforward mathematical definition of phenological lag allows for this method to potentially be applied in different geographic contexts. Where data exists, other researchers can partition the effects of various abiotic forcings on phenological responses that differ from those expected from warming sensitivity alone.Identifying phenological lag, and associated contributing factors, provides a method by which more nuanced predictions of phenological responses to climate change can be made. Thus, this study could improve ecological forecasting models.Weaknesses:The analysis here could be more robust. A more thorough examination of phenological lag would provide stronger evidence that the framework presented has utility. The differences in phenologica lag by study approach, species origin, region, and growth form are interesting, and could be expanded. For example, the authors have the data to explore the relationships between phenological lag and the quantitative variables included in the final model (altitude, latitude, mean annual temperature) and other spatial or temporal variables. This would also provide stronger evidence for the author's claims about potential mechanisms that contribute to phenological lag.

We did examine the relationships of phenological lag with geographic or climatic variables in our analyses. Other than the weak correlations with latitude and altitude cited in the discussion section (lines 292-293), phenological lag was not related to mean annual temperature or long-term precipitation for both leafing and flowering.

The authors include very little data visualizations, and instead report results and model statistics in tables. This is difficult to interpret and may obscure underlying patterns in the data. Including visual representations of variable distributions and between-variable relationships, in addition to model statistics, provides stronger evidence than model statistics alone.

Table 2 shows the influences of geographic or climatic variables, particularly those related to drivers of spring phenology, i.e., budburst temperature, forcing change, and phenological lag, on phenological changes. As quantitative contributions of these drivers have been extracted, the influences of remaining variables are either minor or insignificant. Thus, examination of variable distributions, which has been done in previous syntheses, is probably not necessary.

Some of independent variables were apparently correlated (r <0.6), e.g., MAT with altitude and latitude, budburst temperature with forcing change and spring warming, and forcing change with spring warming.

**Reviewer #3 (Public review):**
Summary:The authors developed a new phenological lag metric and applied this analytical framework to a global dataset to synthesize shifts in spring phenology and assess how abiotic constraints influence spring phenology.Strengths:The dataset developed in this study is extensive, and the phenological lag metric is valuable.Weaknesses:The stability of the method used in this study needs improvement, particularly in the calculation of forcing requirements. In addition, the visualization of the results (such as Table 1) should be enhanced.

Not clear how to improve the calculation of forcing accumulation.

**Recommendations for the authors:**

**Editor (Recommendations for the authors):**
To improve the robustness of the metric and the conclusions drawn, we recommend that the authors:Test the sensitivity of their results to different base temperature thresholds and to nonlinear forcing response models, even for a subset of species. The proposed framework relies on an accurate understanding of species-specific thermal responses, which remain poorly resolved.

Different above-zero base temperatures have been used previously, although justifications are mostly following previous work. As we indicated in our first responses, the use of above-zero base temperatures underestimates forcing from low temperatures, which has more impacts on species with early spring phenology or in areas of cold climate due to greater proportions of forcing accumulations from low temperatures. The use of high base temperatures can lead to an interpretation that early season species require little or no forcing to break buds, which is biologically incorrect. Thus, the use of above-zero base temperatures would be more appropriate for particular locations or species than for meta-analysis across different spring phenology and climatic conditions.

The research on multiple warming is limited in terms of levels of warming used (mostly one and occasionally two) for assessing non-linear forcing responses. This can be the focus of future work.

Our framework is based on drivers of spring phenology and not dependent on “accurate understanding of species-specific thermal responses”. However, a good understanding of species- and site-specific responses to phenological constraints (e.g., insufficient winter chilling, photoperiod, and environmental stresses) does help determine the nature of phenological lag. All these are explained in our paper.

Analyze relationships between phenological lag and additional geographic or climatic gradients already available in the dataset (e.g., latitude, mean annual temperature, interannual variability).

We did examine the relationships of phenological lag with geographic or climatic variables in our analyses. Other than the weak correlations with latitude and altitude cited in the discussion section (lines 292-293), phenological lag was not related to mean annual temperature or long-term precipitation for both leafing and flowering.

Our objective is to understand changes in spring phenology and differences in plant phenological responses across different functional groups or climatic regions, although our approach can be used to study interannual variability of spring phenology. Our metadata are compiled for comparing warmer vs control treatments (often multiyear averages), not for assessing interannual variability.

Improve data visualization to better convey how phenological lag varies across environmental and biological contexts.

See responses above.

Consider incorporating explicit uncertainty estimates around phenological lag calculations. These steps would improve the interpretability and generalizability of the framework, helping it move from a useful heuristic to a more robust and empirically grounded analytical tool.

The calculation of phenological lag is based on drivers of spring phenology with uncertainty depending primarily on uncertainty in phenological observations. Previous uncertainty assessments can be used here (see a few selected studies below).

Alles, G.R., Comba, J.L., Vincent, J.M., Nagai, S. and Schnorr, L.M., 2020. Measuring phenology uncertainty with large scale image processing. Ecological Informatics, 59, p.101109.

Liu G, Chuine I, Denéchère R, Jean F, Dufrêne E, Vincent G, Berveiller D, Delpierre N. Higher sample sizes and observer intercalibration are needed for reliable scoring of leaf phenology in trees. Journal of Ecology. 2021 Jun;109(6):2461-74.

Tang, J., Körner, C., Muraoka, H., Piao, S., Shen, M., Thackeray, S.J. and Yang, X., 2016.Emerging opportunities and challenges in phenology: a review. Ecosphere, 7(8), p.e01436.

Nagai, S., Inoue, T., Ohtsuka, T., Yoshitake, S., Nasahara, K.N. and Saitoh, T.M., 2015. Uncertainties involved in leaf fall phenology detected by digital camera. Ecological Informatics, 30, pp.124-132.